# Planificación nutricional multi-objetivo mediante algoritmos evolutivos

**Javier Quesada-Pajares**
Departamento de Sistemas Informáticos
Universidad Politécnica de Madrid
C/ Alan Turing, s/n, 28031 Madrid
javier.quesada.pajares@alumnos.upm.es

**Cristian Ramirez-Atencia**
Departamento de Sistemas Informáticos
Universidad Politécnica de Madrid
C/ Alan Turing, s/n, 28031 Madrid
cristian.ramirez@upm.es

## Abstract

Este trabajo explora la aplicación de algoritmos evolutivos multi-objetivo en la planificación nutricional, con el objetivo de diseñar dietas personalizadas según las necesidades y preferencias individuales. Mediante computación evolutiva, se genera un plan de comidas semanal que optimiza diversos objetivos nutricionales, como la ingesta calórica y la distribución de macronutrientes, al mismo tiempo que se adapta a restricciones dietéticas del usuario. Se ha utilizado una base de datos con más de 2.500 alimentos, y se ha experimentado con los algoritmos multi-objetivo NSGA-II, SPEA2 y MOEA/D, así como con técnicas de manejo de restricciones basadas en penalización y en separación de los objetivos. Los experimentos comparativos demuestran que el enfoque propuesto cumple eficazmente con las restricciones nutricionales y los requerimientos del usuario, proporcionando un método robusto y adaptable para la planificación de menús.

## 1 Introducción

La nutrición desempeña un papel fundamental en la salud y el bienestar humanos. Una planificación nutricional adecuada es esencial para la prevención de enfermedades crónicas (obesidad, diabetes o trastornos cardiovasculares, entre otros), el mantenimiento de un peso saludable y la optimización del rendimiento físico y cognitivo [1]. A pesar de la creciente disponibilidad de información sobre nutrición, muchas personas enfrentan dificultades para diseñar dietas equilibradas que se ajusten a sus necesidades energéticas, preferencias alimenticias y restricciones dietéticas. Tradicionalmente, la planificación de menús ha sido un proceso manual llevado a cabo por nutricionistas, pero este enfoque puede ser costoso, inflexible y difícil de personalizar a gran escala.

Por otra parte, los enfoques tradicionales para la planificación de dietas a menudo resultan excesivamente simplistas, centrándose frecuentemente en el cumplimiento de recomendaciones nutricionales básicas, sin considerar la naturaleza intrínsecamente multi-objetivo de las decisiones alimentarias en el mundo real. Más allá del mero cumplimiento de recomendaciones nutricionales (como la ingesta adecuada de vitaminas, minerales o fibra), un plan dietético efectivo debe ser también asequible económicamente, adaptable a las preferencias gustativas individuales, culturalmente apropiado y viable en términos de preparación y estilo de vida. Restricciones adicionales, como alergias alimentarias, intolerancias o condiciones médicas específicas, complican aún más el proceso.

Ante esta problemática, los algoritmos evolutivos surgen como una herramienta eficaz para optimizar la planificación nutricional. Estos algoritmos, inspirados en los principios de la evolución biológica, permiten generar combinaciones de alimentos que cumplen con múltiples objetivos nutricionales de manera automática y eficiente. En este estudio, se propone el uso de algoritmos evolutivos multi-objetivo para diseñar planes de alimentación personalizados, ajustados a las necesidades calóricas y la distribución de macronutrientes de cada usuario.

XVI XVI Congreso Español de Metaheurísticas, Algoritmos Evolutivos y Bioinspirados (maeb 2025).

Para ello, se ha desarrollado un sistema que genera menús semanales utilizando una base de datos con más de 2.500 alimentos, garantizando diversidad y precisión en la selección. Se han evaluado distintas técnicas de manejo de restricciones, así como enfoques de optimización multiobjetivo, incluyendo NSGA-II [5], SPEA2 [17] y MOEA/D [16], con el fin de encontrar soluciones óptimas y adaptables.

El resto de este trabajo se organiza como se indica a continuación: la sección 2 introduce el marco teórico sobre algoritmos evolutivos multi-objetivo, así como un breve estado del arte en planificación nutricional. La sección 3 presenta la propuesta de algoritmos evolutivos para planificación nutricional multi-objetivo, presentando la base de datos nutricional usada y definiendo la representación, objetivos y restricciones, así como operadores adaptados de los algoritmos evolutivos. A continuación, la sección 4 muestra los experimentos realizados para comparar tanto los algoritmos evolutivos multi-objetivo como las técnicas de manejo de restricciones para este problema. Finalmente, la sección 5 presenta las conclusiones alcanzadas y algunas líneas de trabajo futuro.

## 2 Marco Teórico y Estado del Arte

### 2.1 Algoritmos Evolutivos

Los Algoritmos Evolutivos (EAs) son una familia de algoritmos de optimización inspirados en la evolución biológica, diseñados para resolver problemas complejos mediante la simulación de procesos evolutivos. Un algoritmo evolutivo típico itera a través de generaciones de soluciones candidatas, aplicando operadores inspirados en la selección natural, el cruce y la mutación para mejorar progresivamente la calidad de las soluciones en relación a uno o varios objetivos definidos.

Dentro del campo de los EAs, se distingue entre problemas de optimización mono-objetivo (SOP) y multi-objetivo (MOP). Mientras que los SOP buscan optimizar una única función objetivo, los MOP, de mayor complejidad y relevancia para problemas del mundo real, requieren la optimización simultánea de múltiples funciones objetivo que a menudo son conflictivas. En los MOP, el objetivo no es encontrar una única solución óptima, sino un conjunto de soluciones Pareto-óptimas que representen diversos compromisos entre los objetivos.

Para abordar los MOP, existen Algoritmos Evolutivos Multi-Objetivo (MOEAs) que incorporan mecanismos específicos para manejar la multi-objetividad y la necesidad de diversidad en el conjunto de soluciones. Entre los MOEAs más destacados se encuentran:

- **Non-dominated Sorting Genetic Algorithm II (NSGA-II) [5]:** Un MOEA que clasifica las soluciones en frentes de Pareto, asignando rangos basados en la no-dominancia. Utiliza el *"crowding distance"* para mantener la diversidad dentro de cada frente, favoreciendo soluciones más aisladas y representativas del espacio de soluciones Pareto-óptimo. Incorpora elitismo para asegurar la retención de las mejores soluciones en cada generación.

- **Strength Pareto Evolutionary Algorithm 2 (SPEA2) [17]:** Otro MOEA popular que asigna a cada solución un valor de fuerza basado en el número de soluciones que domina y una medida de densidad basada en distancia a vecinos más cercanos para promover la diversidad. La combinación de fuerza y densidad en la función de fitness guía la selección de soluciones hacia el frente de Pareto, manteniendo la diversidad de la población.

- **Multi-Objective Evolutionary Algorithm based on Decomposition (MOEA/D) [16]:** MOEA/D es un marco algorítmico que aborda la optimización multi-objetivo mediante la descomposición del problema en un conjunto de subproblemas de optimización escalar más simples, en lugar de tratar directamente el concepto de dominancia de Pareto. Cada subproblema lleva asociado un vector de peso o referencia. Estos subproblemas se optimizan cooperativamente utilizando información de vecindad, lo que permite al algoritmo explorar diferentes regiones del frente de Pareto de manera eficiente y bien distribuida. MOEA/D destaca por su eficiencia computacional y su capacidad para generar frentes de Pareto bien distribuidos, especialmente en problemas con un número elevado de objetivos. La desventaja es que requiere varios hiperparámetros extra: los vectores de referencia iniciales, el número de vecinos a considerar y la probabilidad de compartir información entre vecinos. Para generar los vectores de referencia iniciales hay dos métodos bastante populares: Dan-Dennis, que distribuye uniformemente los vectores de referencia en el espacio objetivo utilizando una partición sistemática; y método incremental, que es una alternativa más flexible que ajusta progresivamente la distribución de las referencias en el espacio objetivo.

Estos algoritmos representan enfoques robustos y ampliamente utilizados para la optimización multi-objetivo, ofreciendo diferentes estrategias para equilibrar la convergencia hacia el frente de Pareto y el mantenimiento de la diversidad de soluciones, esenciales para explorar eficazmente el espacio de soluciones en problemas complejos como la planificación nutricional multi-objetivo.

Por otra parte, para guiar la evolución de los MOEAs hacia soluciones factibles y deseables, se incorporan técnicas de manejo de restricciones [3]. Estas técnicas incluyen la penalización de soluciones no válidas, la reparación de soluciones inviables y el descarte directo de soluciones no factibles. Estas estrategias permiten mantener la diversidad de la población y, simultáneamente, mejorar la convergencia del algoritmo hacia soluciones válidas.

## 2.2 Planificación nutricional

En las últimas dos décadas, se ha producido un auge significativo en la investigación y desarrollo de aplicaciones de algoritmos evolutivos para la planificación nutricional. El trabajo de Kahraman y Seven [8] introdujo un algoritmo genético bi-objetivo para la creación de menús diarios personalizados y saludables, considerando parámetros definidos por el usuario como edad y género, y optimizando la selección de platos para cumplir restricciones nutricionales, minimizar costes y maximizar las preferencias del usuario.

Kaldirim y Köse [9] extendieron este trabajo utilizando el algoritmo multi-objetivo NSGA-II para gestionar de forma separada los objetivos de minimización de coste y maximización de la satisfacción del usuario, mejorando el manejo de restricciones nutricionales. Kashima et al. [10] propusieron una plataforma web para compartir menús generados algorítmicamente, fomentando hábitos alimenticios saludables dentro de una comunidad de usuarios.

Heinonen y Juuso [7] presentaron Nutri-Flow, un software que combina sistemas expertos difusos y algoritmos genéticos para proporcionar guías dietéticas personalizadas, optimizando las recomendaciones alimentarias mediante la evaluación de alimentos y la búsqueda de combinaciones óptimas. Kilicarslan et al. [11] propusieron un modelo híbrido que integra algoritmos genéticos con aprendizaje profundo para la predicción y clasificación de anemias nutricionales, optimizando la planificación nutricional mediante la computación evolutiva. Más recientemente, Joanne B. Cole y Rosita Gabbiannelli [4] exploraron la integración de la IA con el análisis genético para la nutrición personalizada, utilizando algoritmos evolutivos para generar planes de comida y predicciones de salud basados en perfiles genéticos y datos biométricos.

# 3 Algoritmos evolutivos multi-objetivo para planificación nutricional

## 3.1 Selección de la base de datos nutricional

La base de datos de alimentos y platos utilizada para este estudio es "*Composition of foods integrated dataset (CoFID)*" [15], originada a partir del trabajo de Widdowson y McCance, que proporciona información sobre la composición nutricional de alimentos consumidos comúnmente en Reino Unido.

CoFID incluye 2887 alimentos, cada uno de ellos categorizado según su grupo alimenticio. Cada alimento incorpora información sobre una amplia gama de nutrientes, como vitaminas, minerales, azúcares y otros componentes, además de los macronutrientes (carbohidratos, proteínas y grasas). Existen múltiples casos donde un mismo alimento se presenta con diferentes tipos de cocción o de aliño, lo que cambia los valores nutricionales y se considera como dos alimentos o platos distintos. La información necesaria a usar en este problema será el identificador del alimento, la categoría alimenticia a la que pertenece, los gramos de proteína por cada *100g*, los gramos de grasa por cada *100g*, los gramos de carbohidratos por cada *100g* y las kilocalorías del alimento por cada *100g*. En el caso de las bebidas, los valores nutricionales se calculan en base a *100* mililitros. La planificación nutricional se realizará usando estas proporciones.

La base de datos fue preprocesada para eliminar datos incompletos y filtrar categorías de alimentos que no pueden ser consumidos directamente (e.g. *"Grasas y aceites"*), resultando así 2616 registros.

### 3.2 Conceptos nutricionales

En este problema de planificación nutricional multi-objetivo se busca crear un menú equilibrado y personalizado para el usuario, con el propósito de mantener el peso y la masa corporal del individuo.

El primer objetivo es el calórico, siendo necesario conocer cuántas kilocalorías diarias necesita un usuario. La Tasa Metabólica Basal (TMB) es la cantidad de energía que el cuerpo necesita para mantener las funciones vitales en reposo. Existen diversas fórmulas para calcularla, pero en este estudio se hace uso de una de las más extendidas actualmente, la fórmula de Harris-Benedict revisada por Mifflin et al. [13]. Esta fórmula toma en cuenta el peso, la altura, la edad y el sexo del usuario.

$$\text{TMB} = (10 \times \text{peso en kg}) + (6.25 \times \text{altura en cm}) - (5 \times \text{edad en años}) + \begin{cases} 5 & \text{si es hombre} \\ -161 & \text{si es mujer} \end{cases}$$

Como se explica en la definición de la TMB, esta energía es calculada en reposo. Por lo tanto, para alcanzar una aproximación real será necesario multiplicar el resultado por un factor que depende del nivel de actividad física [12]. En este caso, si el hábito del usuario es sedentario, se aplica un factor de 1.2; si es de actividad ligera, un factor de 1.375; si es de actividad moderada, un factor de 1.55, si es de actividad alta, un factor de 1.725; y si es de actividad muy alta, un factor de 1.9. Por ejemplo, para un hombre de *23 años* que pesa *75 kg* y mide *175 cm* su TMB sería de *1734 kcal/día*. Pero como también realiza una actividad moderada, se multiplica por *1.55*, dando un total de *2687.31 kcal/día*.

Otro objetivo es la correcta distribución diaria de macronutrientes (hidratos de carbono, proteínas y grasas). El *Institute of Medicine (IOM)* [14], define los *Acceptable Macronutrient Distribution Ranges (AMDR)* como los rangos de ingesta para macronutrientes que se asocian con un riesgo reducido de enfermedades crónicas y aseguran una ingesta adecuada de nutrientes esenciales:

- $45\% \leq \%$ de carbohidratos $\leq 65\%$
- $10\% \leq \%$ de proteínas $\leq 35\%$
- $20\% \leq \%$ de grasas $\leq 35\%$

Para verificar si la solución respeta estos límites, es necesario calcular cuántas kilocalorías aporta cada macronutriente. Siguiendo el estudio del *IOM*, las kilocalorías por gramo de cada macronutriente serían *4 kcal* de carbohidratos por gramo, *4 kcal* de proteínas por gramo y *9 kcal* de grasas por gramo. La suma de las kilocalorías individuales de cada macronutriente equivale al total de calorías del alimento, dato del que ya se dispone. Con las kilocalorías de cada macronutriente ya calculadas, solo falta dividir estas entre las calorías totales y multiplicarlo por *100* para saber los porcentajes de cada macronutriente. Por ejemplo, para calcular el porcentaje de carbohidratos sería:

$$\left( \frac{\text{kcal de carbohidratos}}{\text{kcal totales}} \right) \times 100$$

### 3.3 Representación del problema mediante algoritmos evolutivos

En este problema, para la estructura del cromosoma usado en el algoritmo evolutivo, se ha planteado que cada gen represente un alimento diferente consumido. Como se muestra en la figura 1, cada día del menú semanal incluye 5 comidas: desayuno, tentempié, almuerzo, merienda y cena. De estas comidas, las tres principales, el desayuno, el almuerzo y la cena, consta cada una de 2 alimentos y 1 bebida, haciendo un total de 3 genes. Las otras dos comidas, consideradas como comidas entre horas, contienen 1 alimento cada una, sumando 2 genes adicionales al cromosoma. Cada día presenta al final 11 genes (o alimentos), por lo que, al tratarse de una planificación semanal, se repite esta cadena por cada día de la semana, dando un cromosoma que presenta un total de 77 genes.

Cada uno de estos genes se encuentra limitado a categorías de alimentos:

- El tercer gen de cada comida principal está destinado a una bebida. De las categorías disponibles, solo se pueden seleccionar *"Bebidas"*, *"Jugos de frutas"*, *"Zumos"* o *"Bebidas Alcohólicas"* (solo si el sujeto es mayor de edad).

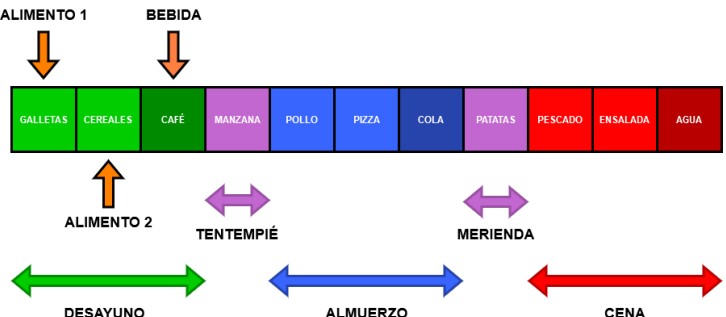

Figure 1: Ejemplo de cromosoma en el problema de planificación nutricional.

- Debido a que la bebida del desayuno suele ser una bebida con leche, café o un zumo, se ha decidido que este gen sea de las categorías *"Leche de vaca"*, *"Bebidas a base de leche"*, *"Bebidas en polvo, esencias e infusiones"*, *"Jugos de frutas"* o *"Zumos"*.

- También los alimentos del desayuno suelen diferenciarse con respecto a los del almuerzo y la cena, que son comúnmente intercambiables entre sí. Por ello, los alimentos de esta comida son del grupo *"Huevos"*, *"Frutas"*, *"Bacon"* o *"Cereales"*.

- Para las comidas entre horas (tentempié y merienda), se seleccionan alimentos de las categorías *"Frutas"* y *"Azúcares"*.

- Para las otras dos comidas principales, el almuerzo y la cena, los alimentos pueden ser de cualquier grupo excepto los de bebidas.

### 3.4 Objetivos y restricciones

El primer objetivo del problema es de las kilocalorías, donde la diferencia entre las kilocalorías que el usuario necesita diariamente y la suma de las kilocalorías de todos los alimentos consumidos en un día sea lo más cercano a 0 posible. La función de evaluación mide cuánto se ha desviado en total en toda la semana, es decir, se va sumando cada desviación de kilocalorías diarias hasta obtener un valor semanal. Para cada usuario, se consultan sus distintos atributos físicos y su nivel de actividad, con los cuales se calcula la tasa metabólica basal y, en última instancia, las kilocalorías.

El objetivo está atado a una restricción *box-constraint*. Se penalizan aquellas desviaciones diarias que superen los límites superior e inferior del 10% respecto a las kilocalorías que el usuario necesita. Busca que las soluciones que se alejen bastante del objetivo sean menos seleccionadas.

$$\text{Minimizar } f_{\text{calorías}}(E_{i,d}) = \sum_{d=1}^{7} \left| k - \sum_{i=1}^{n_d} E_{i,d} \right|$$

$$\text{Sujeto a } 0.9k \leq \sum_{i=1}^{n_d} E_{i,d} \leq 1.1k, \quad \forall d \in \{1, 2, \dots, 7\}$$

donde $E_{i,d}$ es la cantidad de energía (calorías) del alimento $i$ consumido en el día $d$, $k$ es el objetivo calórico diario y $n_d$ es el número total de alimentos consumidos en el día $d$.

Por otro lado, el objetivo relativo a los macronutrientes trata de distribuir correctamente estos a lo largo del día. Se busca que cada tipo de macronutriente se encuentre dentro de los límites recomendados, poniendo como objetivo la media de estos límites. Similar al anterior objetivo, se calcula la diferencia diaria entre estas medias y los valores reales de proteínas, carbohidratos y grasas obtenidos. Posteriormente, la función evalúa la desviación semanal total.

Existe también una restricción *box-constraint* asociada a este objetivo, que exige que la ingesta diaria de cada macronutriente se mantenga dentro de los límites especificados.

$$\text{Minimizar } f_{\text{macronutrientes}} = \sum_{d=1}^{7} \left( \left| \frac{\sum_{i=1}^{n_d} C_{i,d}}{\sum_{i=1}^{n_d} E_{i,d}} - 0.55 \right| + \left| \frac{\sum_{i=1}^{n_d} P_{i,d}}{\sum_{i=1}^{n_d} E_{i,d}} - 0.225 \right| + \left| \frac{\sum_{i=1}^{n_d} G_{i,d}}{\sum_{i=1}^{n_d} E_{i,d}} - 0.275 \right| \right)$$

$$\text{Sujeto a } 0.45 \leq \frac{\sum_{i=1}^{n_d} C_{i,d}}{\sum_{i=1}^{n_d} E_{i,d}} \leq 0.65, \quad 0.10 \leq \frac{\sum_{i=1}^{n_d} P_{i,d}}{\sum_{i=1}^{n_d} E_{i,d}} \leq 0.35, \quad 0.20 \leq \frac{\sum_{i=1}^{n_d} G_{i,d}}{\sum_{i=1}^{n_d} E_{i,d}} \leq 0.35, \quad \forall d \in \{1, 2, \ldots, 7\}$$

donde $C_{i,d}$, $P_{i,d}$ y $G_{i,d}$ son la cantidad de calorías provenientes de carbohidratos, proteinas y grasas, respectivamente, del alimento $i$ consumido en el día $d$.

Otro objetivo a tener en cuenta es la personalización del menú incluyendo los grupos de alimentos favoritos del usuario, o excluyendo aquellos que no le gustan. Se favorecerán los grupos preferidos y se penalizarán los que no. Se puede considerar una restricción *soft constraint*, ya que penaliza las soluciones que la violan pero no las invalida. A diferencia de los otros objetivos, esta función de minimización puede alcanzar valores negativos si existe una gran variedad de alimentos con grupo predilecto en la planificación.

$$\text{Minimizar } f_{\text{preferencia}} = \sum_{i=1}^{N} \begin{cases} -P & \text{si } g(a_i) \in G_+ \\ P & \text{si } g(a_i) \in G_- \\ 0 & \text{en otro caso} \end{cases}$$

donde $N$ es el número total de alimentos consumidos en la semana, $g(a_i)$ es el grupo del alimento $i$ consumido durante la semana, $G_+$ son los grupos de alimento preferidos, $G_-$ son los grupos de alimentos que detesta el usuario y $P$ es el factor de penalización para preferencias.

Finalmente están las restricciones de alergia, que deben penalizar severamente si el usuario es alérgico al grupo del alimento del menú. Al finalizar la semana, se suman todas las penalizaciones de alergias, haciendo que sea inviable para el algoritmo volver a seleccionar un alimento que produzca alergia.

$$\text{Minimizar } f_{\text{alergia}} = \sum_{i=1}^{N} \begin{cases} P_{\text{alergia}}^2 & \text{si } g(a_i) = G_{\text{alergia}} \\ 0 & \text{en otro caso} \end{cases}$$

donde $G_{\text{alergia}}$ es el grupo de alimentos alérgicos del usuario y $P_{\text{alergia}}$ es el factor de penalización.

### 3.5 Operadores adaptados al problema

Para la inicialización y mutación de individuos en el algoritmo evolutivo, se han definido unos operadores que extienden el típico comportamiento de inicialización aleatoria y de mutación uniforme entera para que genere soluciones que sean interesantes en el contexto del proyecto. En vez de seleccionar un índice aleatorio de un alimento de la base de datos, selecciona basándose en las características propias de la solución, explicadas en el apartado 3.3. Es decir, se ha configurado para que se seleccione una bebida cuando el gen sea el tercero de una comida principal o para que los alimentos del desayuno sean de la categoría *"Cereales"*, por ejemplo. Así se consigue que la primera población de la generación cumpla las particularidades del problema.

### 3.6 Manejo de restricciones

En este trabajo se pretenden comparar tres métodos de manejo de restricciones: penalización estática, método separatista y restricciones como objetivos.

En penalización estática se penalizan las soluciones que violan las restricciones. En las funciones que definen las restricciones de calorías y macronutrientes, se calcula una penalización proporcional a la diferencia entre el objetivo y los nutrientes consumidos, multiplicada por una constante de penalización. Tras varias pruebas, se concluyó que la mejor penalización para la restricción de calorías era 50, mientras que para los macronutrientes se usará 30. En el caso de la restricción de alergia, se aplica un factor de penalización si el grupo alérgico aparece en el menú, que se optó por fijar en 100 tras varias pruebas, y elevar al cuadrado el valor de esta restricción. Los valores retornados de las funciones de restricción se suman entre sí, y el resultado se añade a cada uno de los valores objetivo del problema. El algoritmo pasa a calcular 3 objetivos con penalizaciones.

En el método separatista, los objetivos y las restricciones se tratan por separado, sin necesidad de añadir penalizaciones a los objetivos. El algoritmo busca primero soluciones que no violen las restricciones y, cuando las encuentra, busca soluciones que optimicen el problema.

Para el método de restricciones como objetivos, se incorporan las restricciones como nuevos objetivos. La finalidad no es solo encontrar soluciones que cumplan con todas las restricciones, sino también evaluar cuánto se puede mejorar la optimización si se relajan las restricciones. La construcción del método de evaluación se realiza igual que en el método separatista.

# 4   Experimentación

Para la implementación de los distintos métodos evolutivos, se ha usado la librería *Pymoo* [2], especializada en la optimización multi-objetivo. El código está disponible en [1]. Se han creado 5 sujetos de prueba con diferentes características para la experimentación, presentados en la figura 2.

| sujeto_id | peso | altura | edad | sexo | actividad | calorias | gustos | disgustos | alergias |
|---|---|---|---|---|---|---|---|---|---|
| 1 | 78 | 185 | 17 | H | Moderado | 2877.19 | AC, AD, MCA | BR, J, JA, JC, JK, JM, JR | S, SE, SEA, SEC, SN, SNA, SNC |
| 2 | 60 | 170 | 30 | M | Muy Alto | 2567.85 | BAE, FC, FE | C, CA, CD, CDE, CDH | A, AB, AC, AD, AE, AF, AG, AI, AK, AM, AN, AO, AP, AS, AT |
| 3 | 90 | 175 | 40 | H | Alto | 3102.84 | DAP, MAC, SEA | BH, BJS, MIG | PAC, PCA, SNC |
| 4 | 68 | 160 | 55 | M | Ligero | 1710.5 | AF, BNH | MB, QA, QC | F, FA, FC, FE |
| 5 | 72 | 155 | 72 | M | Sedentario | 1401.3 | AM, JC | MG, MR | BA, BAB, BAE, BAH, BAK, BAR, BH |

Figure 2: Sujetos de prueba.

Dado la naturaleza estocástica de los algoritmos, se seleccionaron 31 semillas distintas para garantizar la reproducibilidad de las ejecuciones, y se realizó un análisis de sensibilidad con diferentes valores para cada hiperparámetro de los algoritmos evolutivos (tipos de operadores, probabilidad, tamaño de la población, ...), donde la mejor configuración encontrada se presenta en la tabla 1.

Table 1: Hiperparámetros base del algoritmo evolutivo para planificación nutricional multi-objetivo.

| Hiperparámetro | Valor |
|---|---|
| Tamaño de Población | 100 |
| Número de generaciones | 100 |
| Cruce | Cruce de 2 puntos |
| Probabilidad de cruce | 90% |
| Probabilidad de mutación | 1/77 |

Para comparar conjuntos de resultados se han usado dos métodos. El primero, la tasa de éxito, mide la viabilidad de cada configuración, evaluando el número de soluciones factibles que propone. La mejor configuración es la que mayor tasa de éxito tiene. En segundo lugar, para poder hacer comparaciones en un problema multiobjetivo se ha usado el *hipervolumen*, que evalúa la calidad de un conjunto de soluciones, combinando una medida de la convergencia y diversidad de las soluciones no dominadas. Este mide el volumen de la región del espacio entre las soluciones no dominadas y un punto de referencia. Cuanto mayor sea, mejor será la calidad de las soluciones y la diversidad. Solo se han calculado, dentro de cada una de las 31 ejecuciones de cada experimento, los hipervolúmenes con las soluciones factibles no dominadas encontradas, eliminando las que no cumplen las restricciones. La implementación del hipervolumen de *Pymoo* está basada en la propuesta por Fonseca [6].

Para comparar los resultados finales, se hace uso de tests de significatividad estadística. Para las comparaciones de dos conjuntos de resultados se usará el *test de rangos con signos de Wilcoxon*. Para la la comparación de *n* conjuntos se usa el *test de rangos alineados de Friedman*, con la corrección *post-hoc* de *Shaffer*. En ambos casos, se usará $\alpha = 0.05$ como valor crítico para el p-valor.

## 4.1   Comparación de técnicas de manejo de restricciones

La tabla 2 muestra las tasas de éxito de aplicar las distintas técnicas de manejo de restricciones con el algoritmo NSGA-II, sobre los 5 sujetos de prueba.

---

[1]https://github.com/JaviQuesada/NutritionPlanning

Table 2: Tasa de éxito de los distintos métodos de manejo de restricciones con el algoritmo NSGA-II para planificación nutricional.

| Sujeto | Penalización estática | Método separatista | Restricciones como objetivos |
|---|---|---|---|
| 1 | 99.89% | 93.55% | 0.00% |
| 2 | 97.17% | 97.17% | 0.00% |
| 3 | 100.00% | 100.00% | 0.00% |
| 4 | 100.00% | 100.00% | 0.00% |
| 5 | 100.00% | 100.00% | 0.39% |
| **Promedio** | *99.41%* | 98.14% | 0.08% |

Los resultados del método de restricciones como objetivos resultan muy inferiores, con soluciones no factibles en más del 95% de los casos. Por ello, se va a comparar el hipervolumen únicamente para los métodos separatista y de penalización estática, tal como se muestra en la tabla 3.

Table 3: Hipervolumen (media + desviación típica) de los distintos métodos de manejo de restricciones con el algoritmo NSGA-II para planificación nutricional.

| Sujeto | Penalización estática | Método separatista |
|---|---|---|
| 1 | *104,658,195.91* $\pm$ 36,268,689.23 | 100,247,343.64 $\pm$ 39,747,763.81 |
| 2 | *127,220,326.8* $\pm$ 79,240,038.92 | 124,708,996.15 $\pm$ 56,206,138.41 |
| 3 | *159,596,860.73* $\pm$ 33,309,196.69 | 158,307,003.09 $\pm$ 32,767,547.64 |
| 4 | 174,519,780.85 $\pm$ 26,872,597.24 | *175,681,778.04* $\pm$ 20,850,947.22 |
| 5 | *243,491,393.12* $\pm$ 20,580,336.99 | 237,544,628.36 $\pm$ 25,194,950.15 |
| **Promedio** | *161,497,711.88* | 159,297,149.06 |

La prueba de Wilcoxon sobre estos resultados obtiene un $p-value = 0.1037$, por lo que no se puede asegurar una diferencia significativa entre ambos conjuntos de resultados.

En cuanto a la estabilidad, la penalización estática presenta una tasa de éxito más cercana al 100% en los sujetos 1 y 2 respecto al método separatista, aunque sin significatividad estadística. En cambio, en los tiempos de ejecución destaca el método separatista, que es, de media, 1 segundo más rápido que el otro procedimiento.

Por lo tanto, tanto la penalización estática como el método separatista podrían ser usados para comparar los MOEAs. En este caso, se usará la penalización estática, debido a su ligeramente mejor estabilidad y a que es más fácil de aplicar para el algoritmo MOEA/D.

### 4.2   Comparación de algoritmos evolutivos multi-objetivo

Para MOEA/D, se ha realizado un análisis de sensibilidad de sus parámetros específicos: número de vecinos, probabilidad de compartir información entre vecinos y método de generación de referencias iniciales. Se ha obtenido que los hiperparámetros que obtienen mejores soluciones son 30 vecinos, una probabilidad del 90% de compartir información, y el uso del método incremental con 12 particiones para la generación de referencias iniciales. En el estudio de estos hiperparámetros, se obtuvo al aplicar el test de Friedman con corrección post-hoc de Shaffer que no hay significatividad estadística a la hora de elegir el valor del parámetro probabilidad de compartir información entre vecinos, no obstante se ha elegido el que tiene mejor media entre calidad, estabilidad y velocidad. Por otro lado, el método incremental obtuvo claramente mejores resultados que el método de Das-Dennis en base al test de Wilcoxon. Se ha comprobado que el número de particiones para la generación de las referencias iniciales es un parámetro que, aunque mejora significativamente la calidad de los resultados obtenidos, también aumenta exponencialmente el tiempo de ejecución. Por ello es que se ha decidido tomar una solución compromiso.

Con todas las configuraciones elegidas ya se puede realizar la comparación entre algoritmos. En las tablas 4 y 5 se muestran los resultados obtenidos de la experimentación.

Table 4: Comparación entre MOEAs para planificación nutricional.

| Sujeto | Tasa de éxito | | | Tiempo medio de ejecución | | | Número medio de soluciones no dominadas | | |
|---|---|---|---|---|---|---|---|---|---|
| | NSGA-II | SPEA2 | MOEA/D | NSGA-II | SPEA2 | MOEA/D | NSGA-II | SPEA2 | MOEA/D |
| 1 | 99.89% | 99.21% | *100.00%* | 6.71 | *5.91* | 8.86 | 28.39 | 36.42 | *52.71* |
| 2 | 97.17% | 97.74% | *100.00%* | 6.79 | *5.82* | 8.93 | 29.94 | 23.71 | *55.10* |
| 3 | *100.00%* | *100.00%* | *100.00%* | 6.69 | *5.79* | 7.89 | 34.45 | 34.71 | *56.62* |
| 4 | *100.00%* | *100.00%* | *100.00%* | 6.67 | *5.80* | 8.98 | 37.06 | 40.87 | *50.68* |
| 5 | *100.00%* | *100.00%* | *100.00%* | 6.18 | *6.02* | 9.15 | 56.48 | *67.55* | 59.26 |
| **Promedio** | 99.41% | *5.87* | *100.00%* | 6.61 | *5.87* | 8.76 | 37.26 | 40.25 | *54.87* |

Table 5: Hipervolumen obtenido con los MOEAs para planificación nutricional.

| Sujeto | NSGA-II | SPEA2 | MOEA/D |
|---|---|---|---|
| 1 | $104{,}658{,}195.91 \pm 36{,}268{,}689.23$ | $123{,}175{,}582.95 \pm 37{,}133{,}386.28$ | *$135{,}712{,}971.20 \pm 26{,}445{,}497.37$* |
| 2 | $127{,}220{,}326.8 \pm 79{,}240{,}038.92$ | $121{,}988{,}321.21 \pm 72{,}816{,}991.99$ | *$184{,}266{,}433.84 \pm 33{,}513{,}500.18$* |
| 3 | $159{,}596{,}860.73 \pm 33{,}309{,}196.69$ | $161{,}358{,}829.25 \pm 49{,}426{,}786.12$ | *$175{,}924{,}745.76 \pm 40{,}526{,}295.76$* |
| 4 | $174{,}519{,}780.85 \pm 26{,}872{,}597.24$ | $187{,}220{,}921.73 \pm 24{,}992{,}952.63$ | *$224{,}790{,}089.66 \pm 24{,}538{,}211.84$* |
| 5 | $243{,}491{,}393.12 \pm 20{,}580{,}336.99$ | $245{,}477{,}009.31 \pm 18{,}694{,}107.36$ | *$274{,}223{,}248.84 \pm 36{,}078{,}528.51$* |
| **Promedio** | 161,497,711.88 | 167,044,932.89 | *198,183,897.46* |

Los resultados muestran que MOEA/D presenta una tasa de éxito del 100% para todos los casos, además de obtener el hipervolumen más alto. Aplicando test de Wilcoxon por pares para el hipervolumen, se obtiene que, en efecto, MOEA/D es significativamente mejor que NSGA-II y SPEA2 en cuanto a calidad de las soluciones.

Por otro lado, si miramos a los tiempos de ejecución, se ve claramente que MOEA/D obtiene peor rendimiento que NSGA-II y SPEA2. Entre estos dos últimos, el test de Wilcoxon para los tiempos de ejecución no arroja diferencias significativas entre ellos.

A pesar de su peor rendimiento computacional, el algoritmo MOEA/D resulta la opción más atractiva a usar en estos problemas. El motivo principal es el número de objetivos a optimizar (3), que propicia este tipo de métodos más orientados a la optimización *many-objective* con respecto a otros métodos como NSGA-II o SPEA2 que suelen funcionar mejor con un número menor de objetivos.

## 5    Conclusiones y líneas de investigación

Este trabajo ha demostrado la eficacia de los algoritmos evolutivos multi-objetivo en la planificación nutricional, permitiendo generar menús personalizados que cumplen con restricciones dietéticas y objetivos nutricionales de manera automatizada. A través del uso de algoritmos como NSGA-II, SPEA2 y MOEA/D, se logró optimizar la selección de alimentos considerando múltiples factores, incluyendo la distribución de macronutrientes, las necesidades calóricas y las preferencias individuales.

Los resultados obtenidos evidencian que este enfoque permite una mayor personalización y eficiencia en la planificación de dietas, en comparación con métodos tradicionales. Además, la flexibilidad del algoritmo para adaptarse a diferentes restricciones y preferencias lo convierte en una herramienta potencialmente útil para dietistas, nutricionistas y aplicaciones de salud digital.

Sin embargo, a pesar de los buenos resultados, el sistema presenta ciertas limitaciones. La calidad de los menús generados depende en gran medida de la base de datos de alimentos utilizada y de la precisión de los parámetros nutricionales definidos. Asimismo, el algoritmo puede presentar tiempos de cómputo elevados cuando se trabaja con grandes volúmenes de datos o múltiples objetivos.

Por ello, las líneas de trabajo futuro se centrarán en mejorar la eficiencia del algoritmo, ya sea mediante paralelización o usando técnicas de reducción de dimensionalidad. Por otra parte, en este problema se ha contemplado únicamente 3 alimentos por comida principal y en cantidades de 100 gramos. Para mejorar la personalización de las soluciones, se pretende realizar un enfoque adaptativo que permita seleccionar distinto número de alimentos en cada comida, así como utilizar codificación de números reales para indicar las cantidades de estos alimentos. Finalmente, se pretende expandir la base de datos de alimentos para adaptarla a un mayor número de regiones e incorporar modelos de aprendizaje automático y tratamiento de datos masivos para mejorar la capacidad del sistema para generar dietas más precisas y adaptativas según el historial del usuario.

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
