# OpenReview forum: "Planificación nutricional multi-objetivo mediante algoritmos evolutivos"
_MAEB/2025/Congreso — MAEB 2025_

### Official Review · Reviewer_UHQQ · 2025-03-13
**Este trabajo model como problema de optimización multiobjetivo el diseño de dietas personalizadas según un conjunto de necesidades, preferencias y alergias alimentarias de un perfil de sujeto. El problema consiste en seleccionar, de entre una base de datos de alimentos de los que se conoce su información nutricional, aquellos que conforman una dieta semanal que se define en base a 77 valores (77 variables de decisión del problema), 11 diarios, para desayuno, tentempié, almuerzo, merienda y cena. Hay restricciones sobre los valores que pueden tomar algunos genes, como la distinción entre comida sólida o bebidas. Los objetivos del problema son la minimización de las calorías de la dieta, que ha de estar entre unos valores determinados por la edad, peso, altura y perfil de actividad física del sujeto, maximizar los alimentos preferentes y, finamente, minimizar las posibles incompatibilidades alimentarias (alergias). Se han usado tres algoritmos de la literatura, (NSGA-II, SPEA2 y MOEA/D) para 5 sujetos con diferentes perfiles y se han comparado diferentes técnicas de manejo de restricciones.**

**Rating:** 3
**Confidence:** 5

**Review:**

El trabajo está bien escrito, pero se han identificado las siguientes debilidades:

1.-  Más allá de la complejidad de formular un problema que involucra el modelado de los gustos de personas, la tendencia actual en el dominio nutricional suele ser da los sujetos unas cantidades de tipos de macronutrientes junto con una base de datos (en formato web, fácilmente consultable o app), donde, para cada tipo de comida elige el alimento y la cantidad correspondiente. Generar dietas semanales  sobre una base de datos amplia en la que puede haber alimentos difícilmente conseguible, es complicado.

2.- En la parte técnica, limitar los cromosomas a 77 genes es, quizás, demasiado estricto en cuanto a la variabilidad de los menús que se podrían generar, bien por exceso o por defecto de alimentos en cada comida. Una aproximación con soluciones de longitud variable sería más efectiva y flexible para configurar dietas (e.g., hay slots que se quedan sin usar si un usuario decide no beber nada).

3.- El problema es, en realidad, un problema de satisfacción de restricciones, y así se argumenta la discusión de resultados. Igual la comparación con algún método clásico de ese dominio sería interesante introducir en la comparativa.

4.- En cuanto a los resultados sobre las técnicas de manejo de restricciones, se alcanzan resultados conocidos en la literatura, especialmente cuando se añaden restricciones como objetivos, que hace que su número crezca, y las aproximaciones basadas en optimalidad de Pareto encuentren muchas dificultades. En ese sentido, el NSGA-II debería compararse con MOEA/D en este escenario.

5.- Los autores deben describir en detalle el método usado para calcular el hipervolumen, puesto que es un indicador que requiere normalizar de los frentes antes de calcularlo. También se debe discutir cuántas soluciones factibles se usan en el cálculo de este indicador, y cómo se maneja el escenario en que haya menos de 31 muestras.

6.- Comentarios menores:
	* La sección 2.1 es eliminable, ya que todos esos conceptos son bien conocidos para la audiencia de MAEB.
	* Uso de mayúsculas en las referencias (Moea/d o Spea2)

---

### Official Review · Reviewer_ebka · 2025-03-17
**Problema interesante, comparativa adecuada. Está relacionado con los temas de interés del congreso.**

**Rating:** 4
**Confidence:** 4

**Review:**

Los autores presentan una aplicación de algoritmos evolutivos a un problema práctico. El trabajo es adecuado y relevante para los temas de interés del congreso.

La definición de la representación de las soluciones no permite muchas combinaciones realistas, ya que el cromosoma tiene una longitud fija. En la vida real, una persona puede decidir ayunar por la mañana, comer más de dos alimentos a mediodía (e.g., pollo, arroz, un huevo frito, anacardos y una manzana), etc. Sería más realista representar las soluciones de manera que pudiesen contemplarse comidas con un número variable de alimentos, incluyendo comidas vacías. Quizás una definición de cromosomas lo suficientemente grandes con la posibilidad de introducir genes que representaran un alimento nulo sería suficiente para hacer que el espacio de búsqueda fuese más cercano a la variedad de combinaciones posibles en la vida real. Esto es algo que los autores ya han identificado y que se comenta en líneas de trabajo futuro.

En las tablas de resultados, sería adecuado incluir una fila con los resultados medios por columna. Por ejemplo, en la Tabla 5, sería adecuado incluir una fila al final con el hipervolumen medio que alcanza cada uno de los métodos para todos los sujetos.

Normalmente, se recomienda utilizar NSGA-III en vez de NSGA-II cuando el número de objetivos es alto. Quizás sería interesante incluirlo en la comparación.

Una vez realizado el proceso de búsqueda, sería interesante comparar las soluciones obtenidas con soluciones de un dietista profesional. ¿Qué valores de las funciones objetivos tendrían las soluciones dadas por un dietista profesional para los diferentes individuos?

Es posible que la introducción del usuario (bien sea el usuario final o un dietista elaborando la dieta para otra persona) en el proceso de búsqueda de lugar a mejores soluciones desde el punto de vista de este. Un enfoque interactivo podría ser interesante en este contexto. Véase: *Xin, B., Chen, L., Chen, J., Ishibuchi, H., Hirota, K., & Liu, B. (2018). Interactive multiobjective optimization: A review of the state-of-the-art. IEEE Access, 6, 41256-41279*.

En la mayoría de las ocasiones, las comillas dobles están del revés. O bien las de cierre están puestas como comillas dobles de apertura o viceversa. Ejemplos: líneas 124, 137, 178-179, 180-187, etc.

---

### Official Review · Reviewer_XKZk · 2025-03-19
**Contribución muy interesante en la que se propone el uso de algoritmos evolutivos multiobjetivo par el diseño automático de dietas considerando las restricciones**

**Rating:** 5
**Confidence:** 4

**Review:**

El artículo actual es un trabajo muy completo en el que se estudia el diseño automático de dietas mediante de algoritmos evolutivos multiobjetivo. El modelado del problema está muy bien realizado, siendo muy realista. Se considera una base de datos con un gran volumen de alimentos. El trabajo destaca porque no solo resuelve bien el problema sino que realiza un estudio muy interesante desde el punto de vista de las metaheurísticas al comparar tres algoritmos EMO distintos y tres métodos de modelado de las restricciones existentes. La revisión del estado del arte es muy buena, el trabajo está bien escrito, el estudio experimental es muy completo y los trabajos futuros son muy interesantes.

Solo un pequeño detalle a comentar, más bien para considerarlo en trabajos futuros. Efectivamente, como se comenta en la página 9 ("El motivo principal es el número de objetivos a optimizar (3), que propicia este tipo de métodos más orientados a la optimización many-objective con respecto a otros métodos como NSGA-II o SPEA2 que suelen funcionar mejor con un número menor de objetivos."), es conocido que el rendimiento de NSGA-II se reduce según aumenta el número de objetivos, no siendo recomendable usarlo con más de 3 objetivos. Por esta razón, la decisión de eliminar la tercera alternativa de modelar las restricciones como objetivos en el experimento de la subsección 4.1 puede estar sesgada, ya que solo se considera con el algoritmo EMO NSGA-II. A priori, parece una alternativa interesante para algoritmos EMO como MOEA/D, que no sufren tanto del aumento de la dimensionalidad de los objetivos. Recomiendo estudiar esta alternativa en trabajos futuros.

Un detalle menor, en la línea 298 hay una pequeña errata: "méetodo".

---

### Decision · Program_Chairs · 2025-03-20

Accept